# A Database for Tsunamis and Meteotsunamis in the Adriatic Sea

**Alessandra Maramai \*, Beatriz Brizuela and Laura Graziani**

Istituto Nazionale di Geofisica e Vulcanologia, 00143 Rome, Italy; beatriz.brizuela@ingv.it (B.B.);
laura.graziani@ingv.it (L.G.)

\* Correspondence: alessandra.maramai@ingv.it; Tel.: +39-06-51860210

**Abstract:** In the frame of the Interreg Italy-Croatia program, the EU has funded the PMO-GATE project, focusing on the prevention and mitigation of the socioeconomic impact of natural hazards in the Adriatic region. The Database of Adriatic Tsunamis and Meteotsunamis (DAMT) is one of the deliverables of this project. DAMT is a collection of data documenting both meteotsunami and tsunami effects along the Eastern and Western Adriatic coasts, and it was realized by starting from the available database and catalogues, with the inclusion of new data gained from recent studies, newspapers and websites. For each tsunami and meteotsunami, the database provides an overview of the event and a detailed description of the effects observed at each affected location and gives a picture of the geographical distribution of the effects. The database can be accessed through a GIS WebApp, which allows the user to visualize the georeferenced information on a map. The DAMT WebApp includes three layers: (1) Adriatic Tsunami Sources, (2) Adriatic Tsunami Observation Points and (3) Adriatic Meteotsunamis Observation Points. The database contains 57 observations of tsunami effects related to 27 tsunamis along the Italian, Croatian, Montenegrin and Albanian coasts and 102 observations of meteotsunami effects related to 33 meteotsunamis.

**Keywords:** Adriatic Sea; database; tsunami; meteotsunami; ArcGis; WebApp

## 1. Introduction

Due to the increasing number of extreme events that are being experienced around the world, the interest of the scientific community in natural hazards has grown significantly in recent years, and one of the main targets is the prevention and reduction of risks related to natural events. The Sendai Framework for Disaster Risk Reduction (2015–2030) outlines the overall objectives to substantially reduce disaster risk and losses of lives, livelihoods, and health. It clearly states that in order to diminish the frequency and impact of disasters, it is required to better understand disaster risk (exposure to hazards, vulnerability and capacity and the hazard's characteristics) and furthermore to improve risk governance and increase resilience.

In this regard, many international projects related to the reduction of risk have been funded in the last decade, and the European Union has supported and financed some projects as well. In particular, a wide-ranging cross-border cooperation program between Italy and Croatia called Interreg has been established, focusing on the sea basin, coastal landscapes, green areas and urban areas as well.

The Adriatic Sea is the core center of the Italy-Croatia cooperation area, and it is a joint economic and environmental asset and a natural platform for combined efforts. The coastal area, both in Italy and in Croatia, is exposed to a range of natural hazards, particularly floods, strong winds, drought, earthquakes, tsunamis and meteotsunamis. Taking into account the coastal vulnerability, disaster risk reduction is a critical factor for the social and economic development of the involved countries.

In the framework of the Interreg Italy-Croatia program, Preventing, Managing and Overcoming Natural-Hazards Risks to mitiGATE economic and social impact (PMO-GATE)

has been funded, and it aims at increasing safety from natural and man-made disasters along both the Italian and Croatian coasts of the Adriatic Sea. Making the most of the capitalization on the expertise gained by all partners in previous projects, and using the data and the results available from previous studies, PMO-GATE's purpose is to enhance the level of protection and resilience against natural disasters specific of the region, such as river and sea floods, earthquakes, meteotsunamis and tsunamis. In particular, the project addresses the vulnerability of the Adriatic Sea system with its coasts and islands, implementing cross-border actions in the field of prevention and management regarding the exposure to floods and meteotsunamis in a seismically vulnerable context.

The final outcome of the project is an innovative methodology for preventing, managing and overcoming multi-hazard natural disasters, and climate-induced hazards (such as floods and meteotsunamis) will be combined with non-climate-induced hazards (earthquakes and tsunamis).

PMO-GATE addresses the general public, local and national public authorities, emergency services, education and training centers, universities and research institutes. An effective communication strategy to increase awareness and perception of risk in the population and public authorities is one of the pillars of the project, and the dissemination of results is one of the main goals of the project.

Among the different deliverables of PMO-GATE is the realization of a database for tsunamis and meteotsunamis occurring in the Adriatic region. The Database of Adriatic Tsunamis and Meteotsunamis (DAMT) is the result of the analysis of tsunami and meteotsunami databases existing in the literature for the Adriatic area. Although the two phenomena have different origins, they affect the coasts with similar characteristics and effects, and therefore, in terms of coastal hazard, similar prevention measures must be taken. The two phenomena have always been treated separately, and the creation of a single database is, therefore, important to optimize the hazard and risk studies in the area.

DAMT contains the Adriatic tsunamis present in the Italian Tsunami Effect Database (ITED) [1] and in the Euro-Mediterranean Tsunami Catalogue (EMTC) [2], while as far as meteotsunamis are concerned, the data are essentially those of the Catalogue of Meteorological Tsunamis in Croatian coastal waters (CMTC) [3], integrated with the results of new studies conducted during the PMO-GATE project. Therefore, DAMT is a tool that can be a starting point for a better understanding of the characteristics of such phenomena in the Adriatic area, and it can contribute to hazard and risk assessment for natural events.

PMO-GATE is also a contribution to delivering new data to educational initiatives in the future. In this frame, DAMT is a tool that has also been used for educational purposes during special events addressed to both the general public and scholars, such as World Earth Day [4] and the National Conference in Science Communication 2021 [5], in order to enhance risk perception, which is one of the main aims of the project.

## 2. Tsunamis and Meteotsunamis in the Adriatic Region

Among the various natural hazards to which the Adriatic coasts are exposed, the Istituto Nazionale di Geofisica e Vulcanologia (INGV), as a partner of PMO-GATE, has focussed its efforts on tsunamis and meteotsunamis with the aim of developing and implementing a database of these events: DAMT.

The geographical position of the Adriatic region is peculiar. It lies between the Apennine and Dinaric mountain chains, and it is mostly surrounded by active fold-and-thrust belts and strike-slip faults [6–8]. The presence of microplates and the complex system of faults in the area explain the seismicity which affects the Adriatic region.

Both Italy and Croatia experienced several earthquakes in the past, and a large number of studies were carried out with the aim of better assessing the seismicity of this region. It has been underlined that historical seismicity is poorly documented, and the Adriatic Sea is characterized by quite low seismic activity mainly consisting of earthquakes of moderate magnitudes, but important seismic sequences with main shocks of relevant magnitudes have also been observed [9–16]. Frequent earthquakes occur along the well-known fault

zones, most of which run close to the coastlines or in the open sea and are thus potential sources for tsunamis [17].

In the Adriatic region, the strongest earthquakes (M > 7) have occurred near the eastern margin of the central Adriatic Sea and at the southern end of the basin near the Ionian Islands. The majority of the remaining structures are potentially capable of generating earthquakes of magnitudes $6 \leq M \leq 7$, therefore having significant potential for causing tsunamis [6].

Tsunami is a well-known term mainly used in reference to earthquakes generated ocean waves, and they have a typical period range from a few to tens of minutes to hours. As they approach the coast, these waves can produce severe damage to coastal structures and can cause loss of human life.

Tsunamis are usually generated by submarine earthquakes. However, as the literature demonstrates, in the Adriatic region, as well as along the other Italian coasts, tsunamigenic sources are located both offshore and very close to the coast [1,18]. In this region, a number of tsunamis related to earthquake activity were observed [1,2,18–26]. Most of them had low intensities, but a few events, such as the 1627 Gargano and 1930 Ancona tsunamis, were classified as "very strong".

Italian tsunamis are quite well-documented, and tsunami effects have been observed in many places from the northern to the southern Adriatic region, while in Croatia, tsunami effects have been reported along the northern and southern coasts, with no evidence along the central coast of the country. In Montenegro, tsunamis have been located along the whole coast, and along the Albanian coast, they were mainly located in the area of Valona. Twelve Adriatic tsunamis originated along the Italian coasts, being observed from 1348 to 1930 [18].

Tsunamis generated in the eastern Adriatic region are less frequent; four tsunamis, mainly of low intensities, were ascertained to have occurred since 1667 along the Croatian coast, while the Montenegro coast has been affected by three events since 1667. Six tsunamis have occurred along the Albanian coasts since 1833, all of which were concentrated in the coastal area of Valona. The Croatian coast is much more prone to meteotsunamis, which are meteorologically generated long ocean waves with characteristics similar to those of tsunamis, and they represent a significant hazard for the eastern Adriatic coast [23–25].

Meteotsunamis are formed by storm systems moving rapidly across the water, such as a squall line, and their development depends on several factors such as the intensity, direction and speed of the disturbance as it travels over a water body [23–25]. They can affect localized areas when they reach land. Theb wave heights and spatial extent of meteotsunamis are smaller compared with tsunamis, but they can cause sea level oscillations of several meters and human losses and injuries. Although meteotsunamis are not catastrophic to the extent of major seismically induced events, their occurrence in time and space are higher than those of seismic tsunamis, as the atmospheric disturbances responsible for the generation of meteotsunamis are much more common.

According to recent studies [26], extreme weather has been a common result of the planet's rising temperature, and climate change, like global warming and the sea level rising, may have an impact on the future occurrence and likelihood of meteotsunamis.

The meteotsunami phenomenon is a relatively recent scientific discovery, considering that the first description of tsunami-like effects produced by atmospheric disturbances appeared only in 1931 [27] and that the term "meteotsunamis" was introduced only in 1961 [27].

Thus far, it has been shown that Mediterranean meteotsunamis tend to be stronger in summer. Despite calm conditions at ground level, fast winds of dry air from Africa in the atmosphere 1500 m up seem to trigger atmospheric waves. Mostly during summertime, small-scale strong atmospheric disturbances take place in the Adriatic area, and the coast has a quite complex topography, with a large number of funnel-shaped bays and harbors which have high amplification factors. In fact, the strength of a meteotsunami is also largely dependent on both the topography and the bathymetry of the affected area, and in the

Adriatic Sea, recurrent meteotsunami events are known to strongly impact the lifestyles of the coastal communities, particularly in the Dalmatian islands, where meteotsunamis can generate serious flooding. Several destructive events affected the eastern Adriatic shore in the last few decades, mainly involving Mali Lošinj, Ist, Stari Grad, Vela Luka and Mali Ston. The strongest event hit Vela Luka in Croatia in June 1978, with a wave height of 6 m (crest to trough) and a period of 18 min. This meteotsunami, the most powerful recorded in the Mediterranean, lasted several hours and caused USD 7 million in damage [28,29]. Siroka Bay in the island of Ist and Mali Losinj Bay in the island of Losinj are the two bays in the northern Adriatic where destructive meteotsunamis have occurred recently. Four-meter waves struck Siroka Bay on 22 August 2007, injuring one person and damaging local infrastructure [26]. This locality also experienced meteotsunami effects on 4 October 1984. Mali Losinj Bay was hit on 15 August 2008 by 2-m waves, flooding a forefront of the most populated town of the Adriatic Sea islands and causing a panic during tourist season [30,31].

### 3. The Database of the Adriatic Meteotsunamis and Tsunamis (DAMT)

The understanding of natural phenomena, including tsunamis and meteotsunamis, is essentially based on the deep knowledge of past events. In fact, knowing how many events occurred in the past in a specific region and studying their characteristics help to encourage the study of the more prone coastal areas to assess the hazard and to calibrate models of propagation and inundation. A proper and systematic dissemination of the acquired knowledge is also fundamental to increase the awareness of people living in vulnerable areas.

From this perspective, the availability of a database of events is very important because it is a useful tool for increasing public awareness and, when robust information is available, to validate hazard and risk assessments. In the frame of the PMO-GATE project, one of the deliverables is the creation of the database of Adriatic Tsunamis and Meteotsunamis (DAMT), which was carried out by starting from the catalogs available in the literature, particularlt the Italian Tsunami Effects Database (ITED) [1], the Euro-Mediterranean Tsunami Catalogue [2] and the Catalogue of Meteorological Tsunamis in Croatian coastal waters [3].

DAMT was created by selecting and analyzing the information already available in both of the above-mentioned catalogs, as well as including new data acquired in the frame of the PMO-GATE project from the analysis of recent studies, newspapers and websites. As far as tsunamis are concerned, the data available in the catalogues cover a period of time ranging from 1348 to present day [1,2], while for meteotsunamis, the first information is only available from 1931 onward because for previous events, usually only the year of occurrence was known, and therefore, they were not verifiable or reliable [3]. For each tsunami and meteotsunami observation, the database provides a general description of the event, together with a detailed georeferenced description of the effects observed in each affected location, and offers a complete picture of the geographical distribution of the effects on the coast. In Table 1, the list of tsunamis and meteotsunamis included in DAMT is presented, reporting the main parameters for each event.

The data included in DAMT can be retrieved by the public through a web application similar to that used for ITED [1] which allows the user to visualize the georeferenced information on a map. The WebApp, developed in a freely accessible ESRI ArGis online environment, is accessible through this link (https://ingv.maps.arcgis.com/apps/webappviewer/index.html?id=0f465d51001146d79a6c89884a8e5d8c) (accessed on 20 May 2022) without the need for an Esri user account.

The DAMT WebApp includes three layers: (1) Adriatic Tsunami Sources (ATS), containing the information on the tsunamis that occurred in the Adriatic Basin, (2) Adriatic Tsunami Observation Points (ATOPs), namely the localities where tsunami effects were observed, and (3) Adriatic Meteotsunami Observation Points (AMOPs), which are the localities where meteotsunami effects were observed. In Figure 1, the main screen of the

DAMT WebApp is visible, showing the geographical distribution of the whole dataset contained in the database: tsunamis, where the location coincides with the epicenter of the earthquake that triggered the tsunami (blue squares), tsunami observation points (colored dots in shades of beige) and meteotsunami observation points (red points). Using the layers widget at the top right of the screen, the user can choose the layers to be displayed (*Layer List*).

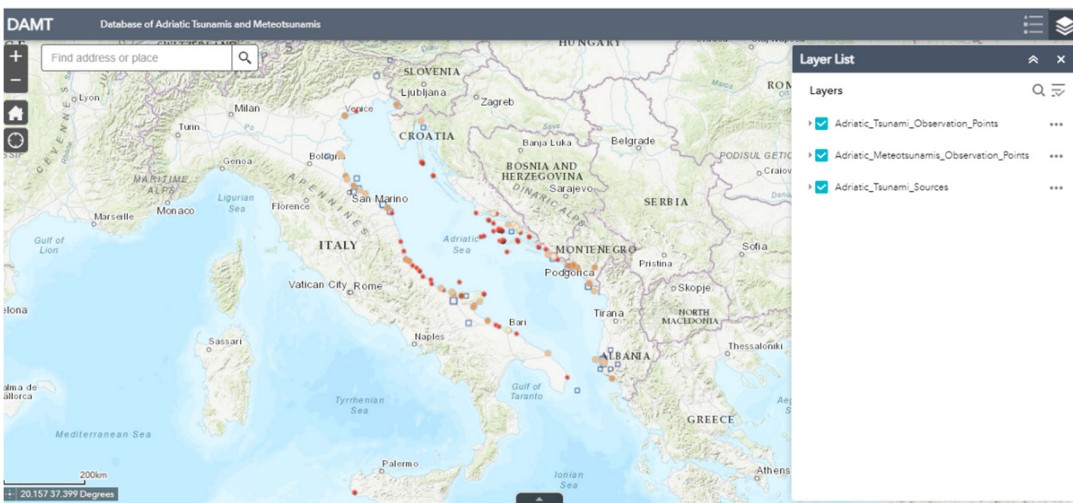

**Figure 1.** Main screen of the DAMT WebApp. On the right, the layer list is shown. Check marks indicate that all levels are highlighted in the map. Blue squares are tsunamigenic sources. Beige-shaded points are tsunami observation points. Red points are meteotsunami observation points.

The data contained in DAMT also populates an attribute table, which can be retrieved through the arrow located at the bottom of the main screen (Figure 2). The tabular information contained in each layer of the WebApp can be exported in csv format, and it can also be filtered by using user-customized expressions in the tab options of the database table (i.e., selecting data from the extent viewed, by date, reliability, cause, region, etc. or by a combination of several of these parameters).

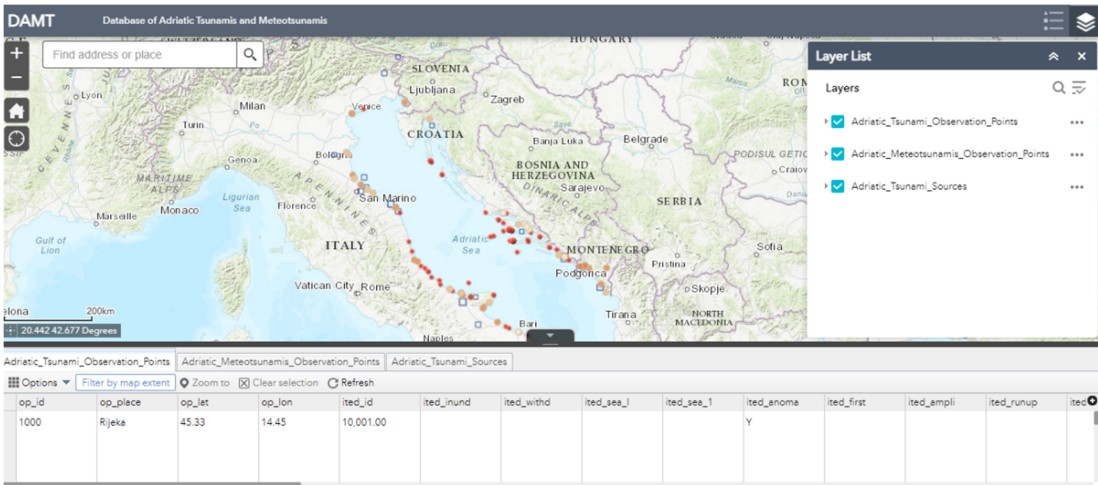

**Figure 2.** The attribute table allows the user to filter the data contained in the layers of the WebApp according to different selections.

As far as tsunamis are concerned, the ATS layer contains the 24 events with the general description of the effects, mainly taken from the EMTC [2].

Twelve tsunamis were located in the Italian coasts, four in the Croatian ones, three in Montenegro and six along the Albanian coasts. By clicking on one of the blue squares on the screen, a pop-up allows the user to obtain general information on the main parameters of the selected tsunami (earthquake parameters and tsunami intensity), and it can link to the general description of the event. In Figure 3a,b, the example of the 30 October 1930 Ancona tsunami is reported.

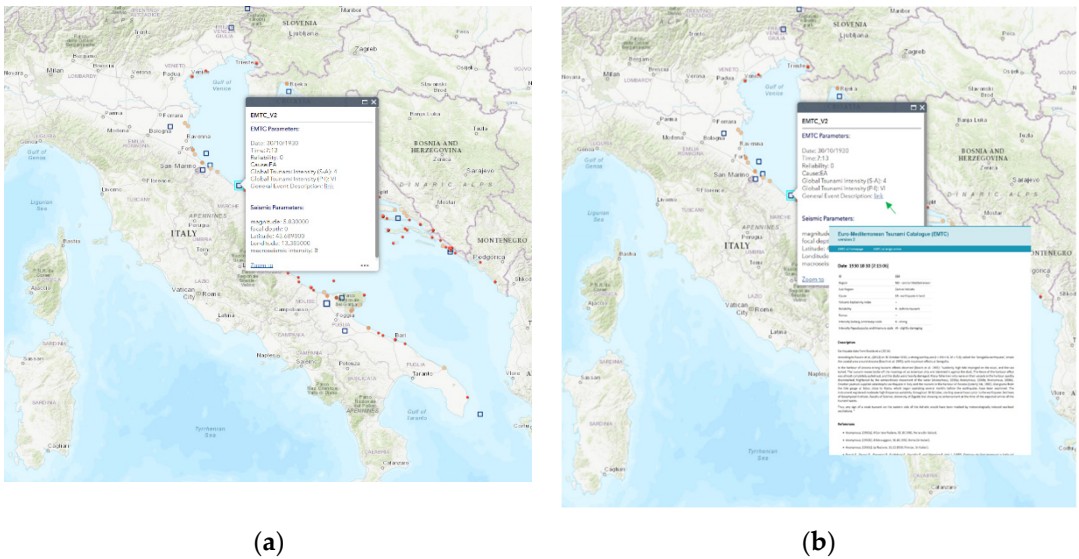

(**a**)                                                                                          (**b**)

**Figure 3.** (**a**) Example of pop-up of the ATS layer, showing the main parameters of the 30 October 1930 Ancona tsunami. (**b**) Example of the 30 October 1930 Ancona tsunami, where the green arrow indicates the link to click on for the general description of the event.

With regard to the Adriatic Tsunami Observation Points (ATOPs) layer, it hosts 57 observation points where tsunami effects were observed: 28 on the Italian coast, 10 in Croatia, 9 in Montenegro and 9 in Albania, respectively (Figure 4a). The Italian ATOPs came from ITED [1], while ATOPs located on the eastern coasts of the Adriatic Sea came from the analysis of the descriptions of the tsunamis contained in EMTC [2]. Additional information has enriched the knowledge on the tsunamis reported in the catalogue. For eastern coast events, additional information was included, and for each observation point, a "local" tsunami intensity value was assessed as "ex novo" on both the Ambraseys–Sieberg [32] and Papadopoulos–Imamura [33] scales. The beige-shaded points are the locations where tsunami effects were observed, while the darker points correspond to more severe effects. For each point, descriptive and, when available, quantitative information (inundation, run-up and wave height values) is provided, along with the corresponding bibliographical references. By clicking on each point, through a pop-up window, the user can obtain information about the effects observed, as well as the main info on the generating tsunami (Figure 4b). Among the Adriatic tsunamis, the maximum run-up observed was 2.5 m at Manfredonia (Apulia region in Italy) during the event on 30 July 1627, while the largest inundation was 50 m at Boka Kotorska (Montenegro) for the event on 15 April 1979, during which one person died [1].

The Adriatic Meteotsunami (AM) layer contains 33 meteotsunami events observed or recorded in 54 places along both the eastern and western Adriatic coasts. Most of the data came from [3], with new information, new events and images being added. Among these 54 places, 8 experienced meteotsunami effects more than once. In particular, from 1931 to the present, Vela Luka has been affected by 18 events, and Stari Grad has been affected by 11 events. In Figure 5, the geographical distribution of the places where meteotsunami effects have been observed is visible. As already mentioned above, meteotsunamis are mainly localized along the coasts of Croatia. As for the northern Adriatic region, although

it is a region prone to Proudman resonances, due to the lack of bays or harbors with high amplification factors, no major meteotsunamis occur in that area [23]. Coversely, the Croatian coast of the central Adriatic, characterized by many islands, channels and narrow bays, is the area where the most meteotsunamis occur (Figure 4), most of which are very powerful. On the Italian coasts, most of the effects were due to the 21 June 1978 event, the strongest in the Mediterranean, but some locations were also affected by the 25 June 2014 event.

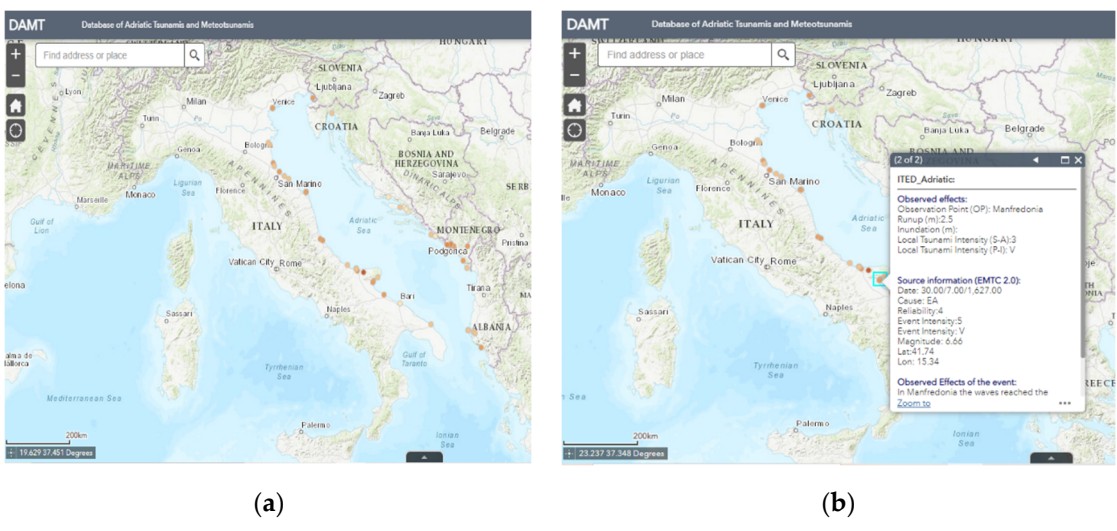

(**a**)                    (**b**)

**Figure 4.** (**a**) Geographical distribution of the Tsunami Observation Points (ATOP layer). (**b**) An example of the pop-up for the Manfredonia (Apulia) observation point.

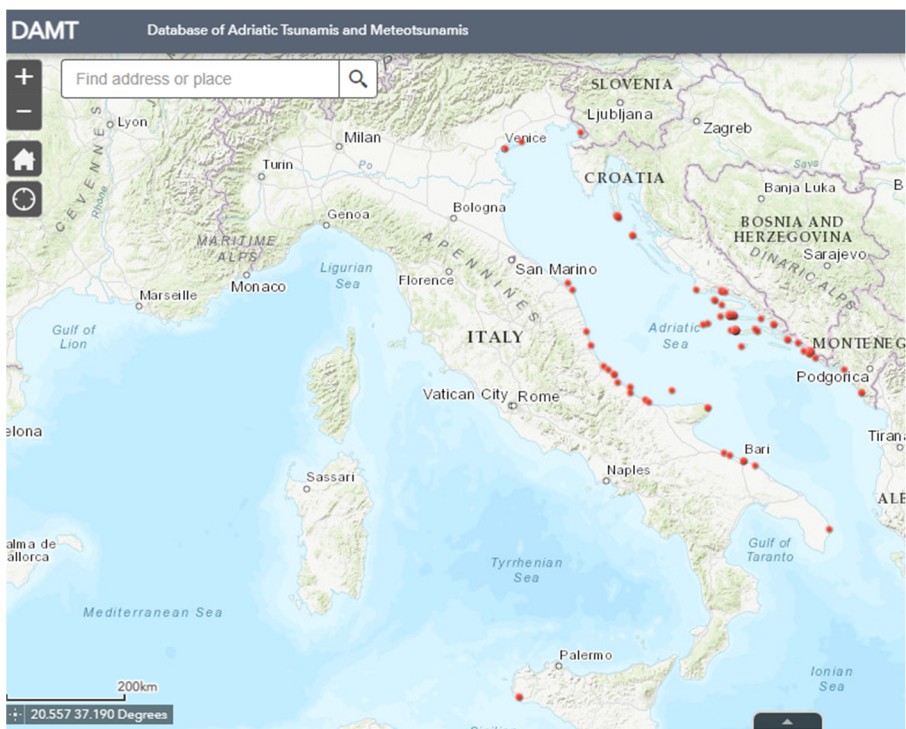

**Figure 5.** Geographical distribution of the meteotsunami effects.

As for tsunamis, and also for meteotsunamis reported in the WebApp, the user can click on each point where the effects were observed. The pop-up window shows the date of the event (or several dates if there is more than one event at the same point), a short description of the observed effects, a link to the general description of the event and the intensity. Since

meteotsunamis are a very complex phenomenon involving meteorological, hydrological and bathymetric factors, for their classification, Orlić and Šepić [3] introduced a value, the QIndex, that indicates how detailed the information about the event is. The QIndex ranges from 1 (elementary description of sea-level variability) to 5 (analysis of oceanographic and meteorological data combined with both oceanographic and meteorological modeling), depending on what kind of bibliographic sources support the available data. In DAMT, the same type of classification has been used, maintaining the QIndex values assigned by [3] and assigning the QIndex to the new events inserted. Figure 6 shows an example of the pop-up for the 11 May 2020 event.

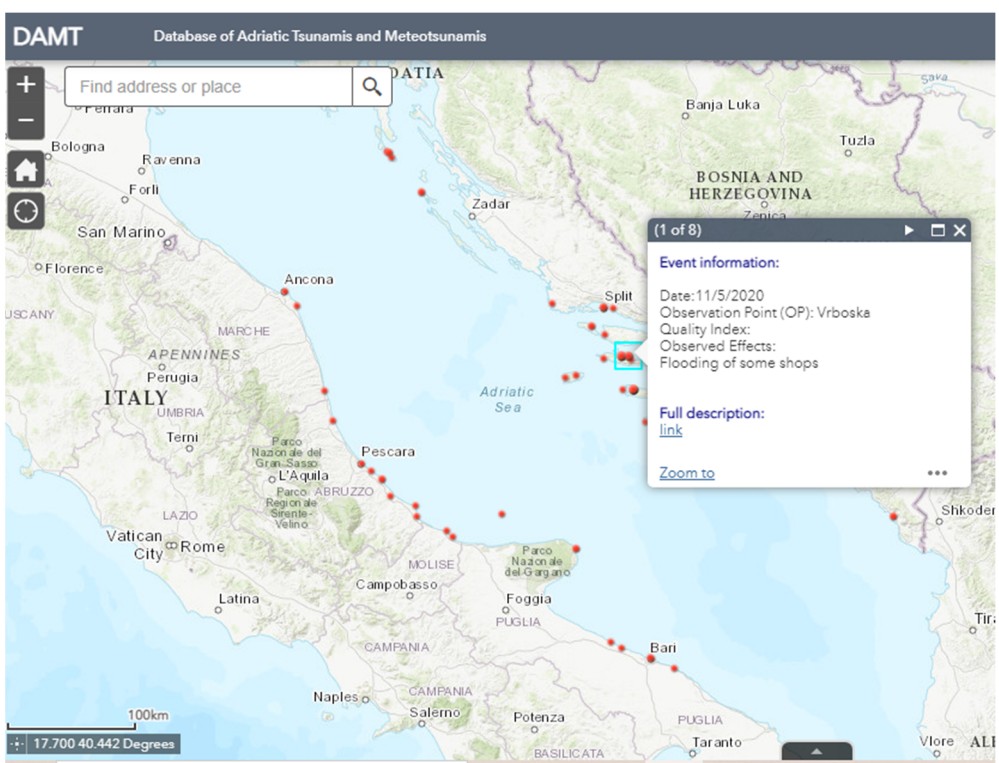

**Figure 6.** Example of a pop-up of the AM layer, showing the 15 May 2020 at Vrboska observation point. By clicking on the full description link, the user can retrieve detailed information on all effects observed at that location over time.

## 4. Discussion

The analysis of the characteristics of past events provides clues to what might happen in the future. In this perspective, a database can be an important and useful starting point to better characterize a region in terms of these phenomena. In the frame of the PMO-GATE project, the realization of a database which includes tsunamis and meteotsunamis in the Adriatic region aims at providing a comprehensive picture of the events in order to highlight that the Adriatic region is prone to these events.

DAMT, for the first time, brings together two phenomena, tsunamis and meteotsunamis, which although originating from different causes produce very similar effects when they hit the coast. Therefore, the measures to be implemented for the prevention and reduction of coastal hazards could be similar.

DAMT was built by starting from the data included in the ITED database [1], in the Euro-Mediterranean Tsunami Catalogue [2] and in the Catalogue of Meteorological Tsunamis in Croatian coastal waters by Orlić and Šepić [3], with the insertion of new events and new info and parameters for the events which had already been catalogued. In Table 1, short descriptions of the events included in the database are reported. DAMT aims to enhance the usability of data and is displayed by means of an ESRI WebApp that allows

the user to query the database in order to retrieve general and detailed information about events, select the events of interest and know the tsunami and meteotsunami history for each observation point. As far as tsunamis, DAMT contains observations of 24 events that have been reported from 1348 to present day, all of which were caused by earthquakes. The analysis of the database puts forth the evidence that in the Adriatic region, the most affected coasts are the Italian ones, which have been affected by 12 events since 1348. Among them, the central northern coast (Emilia Romagna) and the southern one (Apulia) experienced the most relevant events. Four tusnamis have occurred on the coasts of Croatia from 1667 to 1962, three events have affected the coasts of Montenegro in 1667, 1780 and 1979, and finally, the coasts of Alabania experienced six tsunamis from 1833 to 1920.

The tsunamis included in DAMT are classified by a *reliability index*, a value indicating the quality and reliability of the data [21]. Fourteen events (about 60%) have the maximum reliability (four = definite tsunami), and only three events, all of which occurred in Albania in the 1800s, have a reliability of one (very improbable tsunami). Inserting in the database events with low reliability as well allows for keeping a record of those for which little information is available and of which one could otherwise lose track. At the same time, it allows the user to know that the information available for that event is not sufficiently robust to attribute a high reliability value to the event, and therefore, it must be considered with due attention.

Regarding tsunami intensity, to each observation point was assigned a local intensity value based on both the Ambraseys–Sieberg and Papadopoulos–Imamura scales. Most of the events (80%) had intensities that were medium-low, light (2) or rather strong (3), but there were four events with strong (4) and very strong (5) intensities. Two of these tsunamis occurred in Italy in 1627 and 1930, one which took place in Albania in 1920 while the other occurred on the Croatian coast in 1979.

As highlighted by a detailed study of Adriatic tsunamis [34], the analysis of DAMT confirmed that there is no evidence of tsunami effects on the eastern Adriatic side related to the western Adriatc events. However, during two events in the eastern Adriatic region, light effects were observed on the Italian coasts. In particular, after the devastating 1667 Dubrovnik earthquake, anomalous movements were observed in the waters of the canals of Venice, while the tsunami generated by the 1979 Montenegro earthquake was recorded by the tide gauge of Bari in Apulia.

Modeling studies of tsunami propagation from different source areas and tsunami hazards in the Adriatic Sea have shown that earthquakes in the northern Adiatic Sea are not very efficient in generating tsunamis due to the shallow water depth, while the central Adriatic Sea has low seismicity and shallow water, and tsunami waves arriving from the southern Adriatic Sea are partly reflected by the Palagruža rocks [6,34]. Furthermore, the Croatian island chain protects both coasts from tsunami waves propagating from the opposite side of the Adriatic. According to [20], earthquakes located in front of the Montenegrin coast are more efficient at generating tsunamis, and the seismically active region in front of the Albanian coast can generate severe tsunamis, but the modeling results are contradictory as to whether or not tsunamis generated in this area can propagate toward the Italian coasts [6,34]. At the moment, the evidence of the recording of sea level variation during the 1979 Montenegro earthquake in the tide gauge of Bari suggests a possible propagation from the eastern to the western coasts [34]. However, further studies are recommended.

Concerning meteotsunamis, DAMT reports 33 events that occurred from 1931 to the present date, with 103 observations related to 58 places. Compared with tsunamis, for meteotsunamis, the time interval covered by DAMT is much shorter, since the first meteotsunami included occurred in 1931. This is essentially due to two factors: (1) as highlighted in [3], for previous events, usually only the year of occurrence was known, and therefore, they were not verifiable or reliable, and (2) meteotsunamis as a phenomenon are a recent discovery. Just think that the first description of tsunami-like effects produced by

atmospheric disturbances appeared only in 1931 [27], and the term "meteotsunami" was introduced only in 1961 [27].

As also highlighted in [31], for the characterization and proper definition of a meteotsunami event, long-term, high-frequency measurements are necessary. Unfortunately, as mentioned above, historical observations of meteotsunamis in the Adriatic region were extremely scarce and poorly detailed until the early 1930s. In addition, only in recent decades have instruments capable of analyzing such phenomena been installed.

Gathering the information and cataloguing these events is crucial to identify "hotspots", which intensify observational networks in order to obtain, in the future, a robust dataset number to characterize these events, both in terms of effects produced and generated characteristics.

Although the number of events present in DAMT does not allow for performing statistical analysis, it enables us to make general observations.

The exam of the meteotsunami events included in DAMT shows that all of them originated from the Croatian coasts, mainly in the central part of the region. Unlike the tsunamis, the effects of three of them—the 19 September 1977, 21 June 1978 and 25 June 2014 events—were also clearly observed along the Italian coasts. The latter, the strongest meteotsunami in the Mediterranean that was destructive in Vela Luka, affected 14 locations along the Italian coast from Ancona to Otranto, causing serious damage in some places and some injuries in Vieste.

For the classification of meteotsunamis, in the DAMT, we used the criterion proposed by the authors of [3], who assigned a QIndex to each event based on how detailed the bibliographic sources were according to different parameters. In addition, considering the description of meteotsunami effects as if they were tsunami descriptions, in the DAMT, a local intensity value was also assigned for each observation point (OP) according to both Ambraseys–Sieberg [32] and Papadopoulos–Imamura [33] scales. During this process, it was pointed out that the two scales, created for a very similar but nonetheless different phenomenon, are not properly suited for meteotsunami intensity assessment. This is particularly evident for the Papadopoulos–Imamura scale, which has a higher number of degrees (12) with respect to the Ambraseys–Sieberg scale (6 degrees), and the more devastating effects that are described in the higher degrees cannot be produced by meteotsunamis. In fact, even in the case of a very strong meteotsunami, such as the one on 21 June 1978, the observed effects reached a maximum of five on the Ambraseys–Sieberg scale, while they reached a maximum of seven on the Papadopoulos–Imamura scale.

The DAMT contains several reliable data, and it is a tool that can contribute to improving the knowledge of tsunami and meteotsunami activity in the Adriatic area and to increase public awareness along these coasts which, especially in the peak season, are some of most densely populated areas in the Mediterranean. The database has already been used several times for educational purposes to increase the awareness of public authorities, citizens and students toward coastal hazards, and it will be one of the main tools for dissemination of the PMO-GATE project results.

As mentioned earlier, the availability of long-term observations is essential to characterizing the events in a region. Therefore, particularly with regard to meteotsunamis, in the Adriatic area, two factors could be beneficial in the near future: an increase in tide gauge and barometric instrumentation, especially in "hot-spots" that are particularly prone to meteotsunamis on one hand, and on the other hand, in-depth research of bibliographic sources on events that occurred in the past, mainly before 1931. Regarding the latter, a collaboration with Croatian and Albanian researchers is going to start in order to consult local archives and libraries.

**Table 1.** List of the events included in DAMT with the main parameters. Y = year, M = month, D = day of the event, Ev = type of event (T = tsunami; M = meteotsunami), Lat.-Lon. = coordinates of the tsunamigenic earthquake, Description = short description of the main effects, OP = number of observation points where effects were observed; Int. = maximum intensity of the event, where the first number is the value according to the Ambraseys–Sieberg scale and the roman number is the value according to the Papadopoulos–Imamura scale, QI/Rel. = quality index (meteotsunami) and relibility (tsunami), and Source = catalogue from which the data were mainly derived.

| Y | M | D | Ev | Lat. | Lon. | Description | OP | Int. | QI/Rel. | Source |
|---|---|---|---|---|---|---|---|---|---|---|
| 1348 | 1 | 25 | T | 46.500 | 13.580 | Agitation in canals in Venice | 1 | 2/III | 2 | EMTC and ITED |
| 1511 | 3 | 26 | T | 46.210 | 13.220 | Large sea level rise at Trieste | 2 | 3/V | 2 | EMTC and ITED |
| 1624 | 3 | 19 | T | 44.640 | 11.850 | Strong agitation in Po River and coastal lagoons | 3 | 2/III | 4 | EMTC and ITED |
| 1627 | 7 | 30 | T | 41.740 | 15.340 | Large sea withdrawal and flooding in Lesina | 7 | 5/V | 4 | EMTC and ITED |
| 1667 | 4 | 6 | T | 42.600 | 18.100 | Sea withdrawal at Dubrovnik | 3 | 3/V | 4 | EMTC and ITED |
| 1672 | 4 | 14 | T | 43.940 | 12.580 | Sea withdrawal and flooding at Rimini | 1 | 3/IV | 4 | EMTC and ITED |
| 1690 | 12 | 23 | T | 43.550 | 13.590 | Boats stranded in Ancona | 1 | 3/IV | 2 | EMTC and ITED |
| 1731 | 3 | 20 | T | 41.270 | 15.760 | Sea rise at Siponto and Barletta | 2 | 3/IV | 4 | EMTC and ITED |
| 1743 | 2 | 20 | T | 39.850 | 18.770 | Sea withdrawal at Brindisi | 1 | 2/III | 2 | EMTC and ITED |
| 1780 | 9 | 21 | T | 42.400 | 18.500 | Sea withdrawal in Kotor | 2 | 3/IV | 4 | EMTC and ITED |
| 1833 | 1 | 19 | T | 40.400 | 19.900 | Tsunami at Saseno Island | 1 | 2/V | 1 | EMTC and ITED |
| 1838 | 8 | 10 | T | 45.200 | 14.500 | Sea oscillations at Fiume | 1 | 2/III | 2 | EMTC and ITED |
| 1845 | 8 | 16 | T | 42.640 | 18.110 | Sea level rise at Gruž | 2 | 3/IV | 4 | EMTC and ITED |
| 1851 | 10 | 12 | T | 40.700 | 19.700 | Sea level rise at Valona | 1 | 2/IV | 4 | EMTC and ITED |
| 1866 | 1 | 2 | T | 40.300 | 19.400 | Tsunami at Valona and Kanina | 2 | 2/VII | 3 | EMTC and ITED |
| 1866 | 3 | 3 | T | 40.400 | 19.500 | Tsunami at Valona, Kanina and Himara | 3 | 3/IV | 1 | EMTC and ITED |
| 1875 | 3 | 17 | T | 44.210 | 12.660 | Sea flooding at Rimini and Cervia | 5 | 3/IV | 4 | EMTC and ITED |
| 1889 | 12 | 8 | T | 41.830 | 15.690 | Sea agitation in Termoli and Mattinata | 2 | 2/III | 2 | EMTC and ITED |
| 1893 | 6 | 14 | T | 40.300 | 19.700 | Tsunami at Valona | 1 | 2/V | 1 | EMTC and ITED |
| 1916 | 8 | 16 | T | 44.020 | 12.740 | At Tavollo, tsunami waves observed | 1 | 2/IV | 4 | EMTC and ITED |
| 1920 | 12 | 18 | T | 40.500 | 19.500 | Tsunami at Saseno Island | 1 | 5/VIII | 4 | EMTC and ITED |
| 1930 | 10 | 30 | T | 43.690 | 13.380 | Sudden high tide at Ancona | 1 | 4/VI | 4 | EMTC and ITED |
| 1931 | 7 | 21 | M | | | Flooding and ebb at Vela Luka | 1 | 3/IV | 1 | CMCT |
| 1935 | 5 | 28 | M | | | Strong seiches at Vela Luka | 1 | 3/IV | 1 | CMCT |
| 1937 | 9 | 12 | M | | | Large seiches at Vela Luka | 1 | 3/IV | 1 | CMCT |
| 1951 | 11 | 11 | M | | | Strong flood of houses and shops | 4 | 4/VII | 1 | CMTC |
| 1956 | 7 | 21 | M | | | Large flood at Vela Luka | 1 | 2/IV | 1 | CMTC |
| 1962 | 1 | 11 | T | 43.150 | 16.940 | Sea level oscillation at Split | 4 | 2/II | 4 | EMTC and ITED |
| 1966 | 8 | 27 | M | | | Flooding, boats damaged at Korčula | 1 | 4/V | 2 | CMTC |
| 1972 | 2 | 10 | M | | | Flooding of roads at Vela Luka | 1 | 3/IV | 2 | CMTC and DAMT |
| 1977 | 8 | 21 | M | | | Severe flooding, boats damaged at Vela Luka | 1 | 4/V | 2 | CMTC |
| 1977 | 9 | 19 | M | | | Severe flooding, damage, 5 injured at Jesolo | 4 | 4/VI | 2 | CMTC and DAMT |
| 1978 | 6 | 21 | M | | | Inundation, heavy damage, injured people | 30 | 5/VII | 5 | CMTC and DAMT |
| 1978 | 7 | 6 | M | | | Damage on the waterfront at Vela Luka | 1 | 4/V | 2 | CMTC |
| 1979 | 2 | 12 | M | | | Flooding of roads in Vela Luka | 1 | 3/IV | 2 | CMTC |
| 1979 | 4 | 15 | T | 42.020 | 19.070 | Damaging wave at Kotor Bay | 9 | 4/VII | 4 | EMTC and ITED |

**Table 1.** *Cont.*

| Y | M | D | Ev | Lat. | Lon. | Description | OP | Int. | QI/Rel. | Source |
|---|---|---|----|------|------|-------------|----|------|---------|--------|
| 1980 | 7 | 10 | M | | | Strong inundation and heavy damage | 2 | 4/VI | 2 | CMTC and DAMT |
| 1984 | 10 | 5 | M | | | Big waves in Siroka Bay | 1 | 3/IV | 2 | CMTC |
| 1987 | 7 | 26 | M | | | No description | 3 | 1/I | 1 | CMTC |
| 2003 | 6 | 27 | M | | | Large inundation and damage to shops at Stari Grad | 6 | 4/VI | 2 | CMTC and DAMT |
| 2006 | 5 | 24 | M | | | Strong waves at Vrboska | 1 | 2/III | 1 | CMTC |
| 2007 | 8 | 22 | M | | | Damaging wave in Siroka Bay | 2 | 4/V | 2 | CMTC and DAMT |
| 2008 | 8 | 15 | M | | | Flood and damage at Mali Lošinj | 1 | 4/VI | 2 | CMTC and DAMT |
| 2010 | 2 | 19 | M | | | Severe damage to houses and shops at Stari Grad | 1 | 4/VI | 2 | CMTC and DAMT |
| 2014 | 6 | 25 | M | | | Inundation and many boats destroyed | 20 | 4/V | 5 | CMTC and DAMT |
| 2017 | 6 | 28 | M | | | Inundation of the waterfront at Stari Grad | 1 | 3/IV | 2 | CMTC and DAMT |
| 2017 | 6 | 30 | M | | | Inundation of the promenade, slight damage | 2 | 4/V | 2 | CMTC and DAMT |
| 2017 | 7 | 11 | M | | | Flooding of streets and shops | 3 | 4/VI | 2 | CMTC and DAMT |
| 2018 | 3 | 31 | M | | | Flooding of buildings and shops at Stari Grad | 1 | 4/V | 2 | CMTC and DAMT |
| 2018 | 10 | 29 | M | | | Tide gauge record | 1 | 1/II | 2 | CMTC |
| 2019 | 7 | 9 | M | | | Inundation of piers | 2 | 2/II | 2 | DAMT |
| 2019 | 11 | 12 | M | | | Strong flooding of the quays at Vrboska | 1 | 3/III | 2 | DAMT |
| 2020 | 2 | 16 | M | | | Some boats stranded in the Bay of Stobreč | 1 | 3/IV | 2 | DAMT |
| 2020 | 2 | 23 | M | | | No description | 3 | 1/I | 1 | DAMT |
| 2020 | 5 | 11 | M | | | Flooding of shops in Vrboska | 3 | 3/V | 5 | DAMT |
| 2020 | 5 | 14 | M | | | Flooding of the waterfront at Vela Luka | 2 | 3/V | 5 | DAMT |
| 2020 | 5 | 16 | M | | | Flooding of shops and cafes at Vela Luka | 1 | 3/V | 3 | DAMT |

**Author Contributions:** Conceptualization, A.M. and L.G.; methodology, A.M.; software, B.B.; validation, A.M.; formal analysis, A.M.; investigation, A.M.; resources, A.M.; data curation, A.M.; writing—original draft preparation, A.M.; writing—review and editing, A.M., B.B. and L.G.; visualization, A.M.; supervision, A.M.; project administration, A.M.; funding acquisition, A.M. All authors have read and agreed to the published version of the manuscript.

**Funding:** This research was funded by the EUROPEAN UNION, Programme Interreg Italy-Croatia, Project "Preventing, managing and overcoming natural-hazards risks to mitigate economic and social impact" PMO-GATE ID 10046122.

**Institutional Review Board Statement:** Not applicable.

**Informed Consent Statement:** Not applicable.

**Data Availability Statement:** The datasets presented in this study can be found in online repositories. The name of the repository and accession number can be found below: https://ingv.maps.arcgis.com/apps/webappviewer/index.html?id=0f465d51001146d79a6c89884a8e5d8c (accessed on 20 May 2022).

**Acknowledgments:** The authors thank Mario Locati, (INGV, Milan) for the technical support during the realization of the DAMT database.

**Conflicts of Interest:** The authors declare no conflict of interest.

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
