# Peer review of "A Database for Tsunamis and Meteotsunamis in the Adriatic Sea"

_applsci, doi:10.3390/app12115577_

Round 1

Reviewer 1 Report

The manuscript titled "A database for tsunami and meteotsunami in the Adriatic Sea" presents a web application for accessing and using the GIS data.

Although the manuscript is well written and the derived application could be useful to the general public, there is no significant scientific contribution.

Author Response

Although the manuscript is well written and the derived application could be useful to the general public, there is no significant scientific contribution.

The paper has been improved, from the introduction to the discussion. A table showing the events included in the DAMT has been also added to the paper. As regard the "scientific contribution", the PMO-GATE project aims to capitalise on the results of previous projects by using existing data as much as possible. The DAMT is a new and innovative database that, for the first time, with detailed and reliable data, gather tsunamis and meteotsunamis, two phenomena that have similarities with regards to the methodologies to be implemented for coastal hazard prevention and mitigation.  It collects data from both tsunami and meteotsunami catalogue available in literature, deeply revised and with the insertion of new parameters and new events. The DAMT database, like all database, does not have the claim to present innovative scientific results, but it aims at being a tool designed and built to easily provide tsunami and meteotsunami data that can be used both to validate hazard studies and to disseminate results to general public, local authorities and schools, as already tested in several contexts. 

Reviewer 2 Report

The submission is a mature manuscript dedicated to the presentation of a new database for monitoring tsunamis and meteotsunamis in the Adriatic Sea. While I generally think that this paper would well contribute to international research debates on monitoring natural hazards and vulnerability, there are just some minor points I would like to suggest for a revision round:

  • The important discussion section could be expanded. In its current version, it includes several references to pevious projects related to the Adriatic Sea. Of course, this is the (geospatial) case study. However, when developing a new database, this could also be inspired by similar projects in the world. What are the advantages or disadvantages of the new database over well-known similar projects? This increased transparency could help the readers to get a broader picture of the impact of the project.

  • Geohazards are a topic which are also very important in education. Some studies identified a clear lack of student’s knowledge (e.g. https://www.researchgate.net/profile/Theofilos-Toulkeridis/publication/341273560_TSUNAMI_HAZARDS_IN_ECUADOR_-_REGIONAL_DIFFERENCES_IN_THE_KNOWLEDGE_OF_ECUADORIAN_HIGH-SCHOOL_STUDENTS/links/5eb72b7092851cd50da3c839/TSUNAMI-HAZARDS-IN-ECUADOR-REGIONAL-DIFFERENCES-IN-THE-KNOWLEDGE-OF-ECUADORIAN-HIGH-SCHOOL-STUDENTS.pdf & https://doi.org/10.1016/j.jvolgeores.2007.12.010. Your project could be a contribution to deliver new data to educational initiatives in the future.

Author Response

The important discussion section could be expanded. In its current version, it includes several references to pevious projects related to the Adriatic Sea. Of course, this is the (geospatial) case study. However, when developing a new database, this could also be inspired by similar projects in the world. What are the advantages or disadvantages of the new database over well-known similar projects? This increased transparency could help the readers to get a broader picture of the impact of the project. Geohazards are a topic which are also very important in education. Some studies identified a clear lack of student’s knowledge (e.g. https://www.researchgate.net/profile/Theofilos-Toulkeridis/publication/341273560_TSUNAMI_HAZARDS_IN_ECUADOR_-_REGIONAL_DIFFERENCES_IN_THE_KNOWLEDGE_OF_ECUADORIAN_HIGH-SCHOOL_STUDENTS/links/5eb72b7092851cd50da3c839/TSUNAMI-HAZARDS-IN-ECUADOR-REGIONAL-DIFFERENCES-IN-THE-KNOWLEDGE-OF-ECUADORIAN-HIGH-SCHOOL-STUDENTS.pdf & https://doi.org/10.1016/j.jvolgeores.2007.12.010. Your project could be a contribution to deliver new data to educational initiatives in the future.

         REPLY: We would like to thank the referee for the interesting and useful suggestions, which we gladly accepted, especially regarding the importance of the database in the dissemination of the results. The paper has been improved, from the introduction to the discussion, with important amendments to the text and with the introduction of a table showing the events included in the database with the main information. It was pointed out that one of the goals of the PMO-GATE project is to use as much as possible data gained in previous projects and it was underlined that the database is a versatile and useful tool for the dissemination of results to general public, local authorities and schools, as already tested in several occasions.

Reviewer 3 Report

This manuscript described the development of a database for tsunamis and meteotsunami in the Adriatic Sea. The database can be accessed through a GIS WebApp, which allows the user to visualize the georeferenced information on a map. I am teaching WebGIS with a working ArcGIS account and still couldn't get access to the App. Please make sure the App is accessible when resubmitting. 

In addition, the similarity of session 1-3 is high. Please find the attached similarity report.

Author Response

Comment 1. This manuscript described the development of a database for tsunamis and meteotsunami in the Adriatic Sea. The database can be accessed through a GIS WebApp, which allows the user to visualize the georeferenced information on a map. I am teaching WebGIS with a working ArcGIS account and still couldn't get access to the App. Please make sure the App is accessible when resubmitting. 

REPLY: We would like to thank the referee for pointing out the error in accessing the WebApp. The correction has been done and everything should now run properly. 

Comment 2: In addition, the similarity of session 1-3 is high. Please find the attached similarity report.

REPLY: As concern the similarities, they are mainly due to the fact that one of the goals of the PMO-GATE project is to use as much as possible data gained in previous projects and this, in the description of the database, obliges us to comply with what was presented in previous projects. Furthermore, both EMTC and ITED database, from which most of the tsunami data included in the new DAMT database are taken, were created by ourselves. The DAMT database has been built along the lines of EMTC and ITED and it has been modified to include the new phenomenon, meteotsunamis. Therefore, this causes similarities between the two sessions. However, the paper has been improved with the introduction of new parts of text and a table showing the list of events included in the database, with the main information. 

Reviewer 4 Report

The authors have done a very good work. I would like to suggest to add the impact of earthquakes in Himalayan region and around may be added to sound the paper more accurately.

Author Response

Comment: The authors have done a very good work. I would like to suggest to add the impact of earthquakes in Himalayan region and around may be added to sound the paper more accurately.

Reply: We would like to thank the referee very much for the appreciation of the work. Probably there is a part in the comment which is related to another paper, since the Himalayan region is not in the area of interest of our paper. 

Round 2

Reviewer 1 Report

The addition of the table containing all the data from the dataset did not improve the quality of the manuscript, but some of the other changes did. Nevertheless, the manuscript is well written and makes a useful web application. 

In my opinion, the proposed dataset does not represent a scientific contribution, especially since most of the data presented are only from different projects. 

The authors should present an example of how to use the data from the proposed dataset, as is common when a dataset is proposed. Along with the proposed use, the statistical analysis of the generated results should be added.

Author Response

Reply:

As we have added in the paper, the number of the data in the database does not allow to perform significant statistical analysis. Therefore, we gave a general description of where the events occur in the Adriatic region and what are the effects observed. This is the main purpose of a database. This qualitative description is the first step towards a deeper understanding of the phenomenon. A more detailed and deep knowledge can only be achieved through long-term observational data.

The WebApp, in addition to other possibilities, it also allow the user to explore both the tsunami and the meteotsunami history for each observation point.  

Reviewer 3 Report

https://ingv.maps.arcgis.com/apps/webappviewer/index.html?id=0f465d51001146d79a6%20225%20c89884a8e5d8c

The link is still not working in the newer version, please double check

"Item does not exist or is inaccessible."

Author Response

Dear referee,

I am very sorry for the problem  you had with the link again. I don't know what could have happened. We have checked again and it is working properly. We also asked other colleagues to try and they were able to access it without any problems. I will send you the work again, hoping there won't be any problem. thanks
